# Anti-Inflammatory Property Establishment of Fulvic Acid Transdermal Patch in Animal Model

**Maria A. Konnova** [1], **Alexander A. Volkov** [1], **Anna G. Solovyeva** [2,3], **Peter V. Peretyagin** [2] **and Nina B. Melnikova** [4,*]

1 Department of Pharmaceutical Chemistry, Privolzhsky Research Medical University, 10/1 Minin Sq., 603950 Nizhny Novgorod, Russia; mash.konnova@yandex.ru (M.A.K.); volkov-aj@yandex.ru (A.A.V.)
2 Department of Experimental Medicine, Privolzhsky Research Medical University, 10/1 Minin Sq., 603950 Nizhny Novgorod, Russia; sannag5@mail.ru (A.G.S.); peretyaginpv@gmail.com (P.V.P.)
3 Department of Biochemistry and Biotechnology, Lobachevsky State University, 23 Gagarin Av., 603022 Nizhny Novgorod, Russia
4 Faculty of Chemistry, Lobachevsky State University, 23 Gagarin Av., 603022 Nizhny Novgorod, Russia
* Correspondence: melnikovanb@gmail.com; Tel.: +7-902-309-2298

**Abstract:** The formulation of the transdermal patch with fulvic acid (FA) on an emulsion basis using pluronic Kolliphor® p237 as a permeability enhancer was developed and studied for anti-inflammatory properties. FA was isolated from the peat in the Nizhny Novgorod region of Russia and characterized as a potential active pharmaceutical ingredient. In vitro studies of the release of FA from the transdermal patch, as well as the FA penetration through the acetyl cellulose membrane using the Franz diffusion cell, showed its high efficiency (56% and 90%, respectively, in 8 h). In the in vivo experiment, qualitative and quantitative features of the rat knee caused by complete Freund's adjuvant-induced arthritis (morphological changes, the FA influence on the biochemical indexes) were studied. The inflammatory process that developed within 15 days was accompanied by the activation of antioxidant oxidoreductase enzymes (by 50–70%), the increase in the cross-sectional diameter of the cartilage, and the increase in the values of marker indicators of the process of rheumatoid arthritis. Within 7 days of treatment, under the influence of FA, the values of ESR, RF, leukocytes, C-reactive protein, as well as the biochemical parameters characterizing oxidative stress (SOD, catalase, glutathione reductase, LDH, G6PD) normalized, and the edema reduced. These results may be useful for arthritis treatment using the transdermal patch with FA.

**Keywords:** fulvic acid; humic substances; transdermal patch; release and permeability; complete Freund's adjuvant-induced arthritis; oxidoreductase

## 1. Introduction

Fulvic acids are natural compounds with phenolic, acidic and other groups, and they are of great interest to medicine. Despite the fact that this group of substances has a different composition, depending on the source of extraction (silt, soil, peat, lignins, and others), they have common antioxidant [1–7], antitumor [8–11], antiviral [12–14], anti-inflammatory [3,12,15–18] properties.

The most studied of them is the anti-inflammatory effect of FA. The experiment on rats showed a decrease in paw edema in rats in inflammation caused by carrageenan [3]. FA reduces the level of C-reactive protein in patients with osteoarthritis of the knee, accompanied by progressive cartilage destruction [19]. Like all humic substances, FA has a dose-dependent effect on the immune system, both inducing and reducing inflammatory processes [20].

Rheumatoid arthritis (RA) is a disease that can be associated with overactive immune cells causing inflammation. The RA proceeds against the background of an increase in the process of oxidative stress. During pathogenesis, ROS generation is activated. ROS products have a destructive effect on biological molecules, initiate lipid peroxidation (LPO),

and also lead to morphological and functional disorders of cellular structures, degradation of collagen, damage to connective tissue [21,22]. The damaging LPO process is counteracted by the antioxidant system.

In this regard, it is very important to study FA, which has antioxidant activity, as a potential protector of oxidative stress. FA as a polyphenolic compound is capable of being a chelator of metal ions of variable valence [23–26], acting as a scavenger for ROS, and being a reducer of oxidized forms of other antioxidants.

Despite the high need for low-toxic anti-inflammatory substances of natural origin for the treatment of RA, dosage forms with FA are practically not discussed in the literature. There are scattered data about FA gels. For example, interesting studies have appeared on FA and thymoquinone containing nanogels for the treatment of psoriasis [27,28]. There are some studies on fulvic acid gels for the treatment of burns [29] and eczema [30]. Also, during experiments in Korea, an analysis of the effectiveness of a cream containing fulvic acid on various indicators of human skin was performed [31]. Despite the many positive effects shown by FA in gels, this dosage form is not without drawbacks. In particular, it is necessary to introduce preservatives and stabilizers into the composition of gels, which prevent the rapid oxidation of FA as a phenolic compound during storage and use. Secondly, FA in solutions is capable of aggregation and the formation of larger particles, which also reduces the stability of the gel. Transdermal patches can be a good alternative to gels, in which FA is more stable during storage, since it is not prone to oxidation and aggregation in the absence of a solvent. Moreover, the patch form is more convenient in the treatment of chronic inflammatory diseases such as RA [32,33].

The complete Freund's adjuvant-induced arthritis of knee model may be a simple model of RA [34], which makes it possible to compare the effect of FA on the course of the RA process with the effect of other drugs on oxidative stress [35].

The aim of the present paper was to study the effect of FA in the transdermal patch on biochemical indexes and morphological changes in the treatment of inflammation caused by complete Freund's adjuvant-induced arthritis of knee in rats. We have investigated: (i) an anti-inflammatory formulation of the transdermal patch with FA isolated from the peat in the Nizhny Novgorod region of Russia; (ii) the release of FA from the transdermal patch; (iii) changes in the marker and biochemical parameters (erythrocyte sedimentation rate (ESR), leukocytes, C-reactive protein, rheumatoid factor (RF)). Activity of antioxidant enzymes (superoxide dismutase (SOD), catalase, glutathione reductase (GR), glucose-6-phosphate dehydrogenase (G6PD), lactate dehydrogenase (LDH), malondialdehyde (MDA) level, aldehyde dehydrogenase (ALDH) activity) were assessed during treatment with the transdermal patch. In addition, the assessment of the histopathological state of the knee joint before and after treatment was made.

## 2. Materials and Methods

### 2.1. Materials

Lowland Peat of Nizhny Novgorod region (Russia) was received from LLC "ES-SON" (Nizhny Novgorod, Russia). In the study, we used sodium hydroxide (99.0% purity, Bashkir Soda Company JSC, Sterlitamak, Russia), sulfuric acid (99.9% purity, EKOS-1, Staraya Kupavna, Russia), Superlite™ DAX-8 (Supelco—Sigma-Aldrich Co., Bellefonte, PA, USA), Cationite KU 2–8 H+ (NevaReactiv, Saint Petersburg, Russia), Kolliphor® 237 (block copolymers of ethylene oxide ($n$ = 64) and propylene oxide ($n$ = 37), BASF Pharmaceuticals, Ludwigshafen, Germany), Kolliphor® 338 (block copolymers of ethylene oxide ($n$ = 141) and propylene oxide ($n$ = 44), BASF Pharmaceuticals, Ludwigshafen, Germany), xantan gum (Sisco Research Laboratories Pvt. Ltd., Mumbai, India), polyethylene oxide-400 (PEO-400, Impexpro, Kazan, Russia), polyethylene oxide-1500 (PEO-1500, LDhim, Dzerzhinsk, Russia), propane-1,2,3-triol (glycerin, 99.9% purity, Reachem, Staraya Kupavna, Russia), poly [1-(2-oxo-1-pyrrolidinyl)ethylen] (polyvinylpyrrolidone K-17, PVP K-17, PLASDONE™ K-17, Ashland industries, Rotterdam, The Netherlands), (2S,3R,4R,5R)-Hexane-1,2,3,4,5,6-hexol (sorbitol, C Pharm™ Sorbitex P, Cargill Inc., Minnetonka, MN, USA), dimethyl sulfoxide (DMSO,

99.9% purity, EKOS-1, Staraya Kupavna, Russia), polyoxyethylene (80) sorbitan monooleate (Tween-80, HimPiterTorg, Saint Petersburg, Russia), emulsion wax (Rusoleochem, Vyazma, Russia), regenerated cellulose membrane (0.45 μm pore size, 47 mm circle, Water Laboratory, Saint Petersburg, Russia), cellulose acetate membrane (OE 67, 0.45 μm pore size, 25 mm circle, Cytiva Whatman™, Little Chalfont, Buckinghamshire, UK).

### 2.2. Solid-State $^{13}C$ NMR

Registration of solid-state $^{13}C$ NMR spectra was performed by JNM-ECX400 spectrometer (JEOL, Tokyo, Japan–9.39 T, 100.5 MHz) in the solid phase at room temperature using the technique of cross-polarization and magic angle rotation (combined CP-MAS experiment) with frequency 10 kHz rotation in 4 mm zirconia rotors. The magic angle of sample rotation (MAS) was determined at a rotation speed of 6 kHz using a KBr standard sample. The CW (Continuous Wave) decoupling method was used. VACP (variable amplitude cross-polarization) and RAMP-CP method were used. Adamantane (29.5 ppm) was used as a reference sample; the resolution was also adjusted using it. All CP-MAS experiments were performed at room temperature; proton decoupling was performed using double phase pulse modulation (TPPM). The duration of the 90° pulse for $^{13}C$ nuclei was 2.93 μs and was determined using a sample of hexamethylbenzene. The total number of scans was 4096.

The spectra were recorded using the Delta 4.3.6 program and processed using the ACD/NMR Processor Academic Edition, Ver. 12.01.

### 2.3. FTIR Analysis

FTIR spectra were obtained in the 400–4000 $cm^{-1}$ range by an IR Prestige-21 FTIR spectrometer (Shimadzu, Kyoto, Japan). The resolution was 0.5 $cm^{-1}$, and the number of scans was 45.

### 2.4. Photoluminescence Analysis

Fluorescent spectra were obtained using spectrofluorimeter CM 2203 (Solar, Minsk, Belarus). One-dimensional emission spectra were taken in the wavelength range of 400–550 nm at a constant wavelength of 360 nm. Excitation spectra were recorded in the range of 300–500 nm at a fixed radiation wavelength of 520 nm.

### 2.5. SEM and EDXMA Studies

The samples were visualized by scanning electron microscopy (SEM) using a JSMIT300LV (JEOL, Tokyo, Japan) microscope with an electron beam diameter of about 5 nm and a probe current below 0.5 nA (operating voltage 20 kV). The surface topography of the powders was studied using low-energy secondary electrons and backscattered electrons. The elemental composition of the powders was studied using X-ray microprobe analysis (XRM) with an X-MaxN 20 detector (Oxford Instruments, Oxfordshire, UK).

### 2.6. Zeta Potential Analysis

Zeta potential was measured with Zetacheck particle charge reader (LLC "Microtrak", Saint Petersburg, Russia).

### 2.7. The Particle Size Analysis

Particle size distribution in the solutions was determined by dynamic light scattering using Nanowin particle size analyzer (LLC "Microtrak", Saint Petersburg, Russia).

### 2.8. Fulvic Acid Preparation

The raw material for FA obtaining was the peat containing humin (40–50%), humic acids (~30%), and fulvic acids (10%). In addition, the peat contains other organic substances and inorganic impurities. FA was obtained by the following method. Briefly, the peat (200 g) was US-alkaline hydrolyzed for 1 h at 80 °C, and then after humin was removed by centrifugation, the final solution was acidificated by $H_2SO_4$ up to pH 1–2, and humic acid

was removed. Purification of the FA solution was carried out by the Lamar method [36]. The drying of the FA solution was carried out by freeze-drying (−80 to −40 °C for 8 h–LGJ-10 (−80 °C), Vikumer: A Brand of Vekuma Machinery Co., Ltd., Beijing, China).

### 2.9. Transdermal Patches Preparation

Basic composition: (a) PVP K-17 (0.5 g), PEO-400 (0.02 g), and PEO-1500 (0.2 g) were added in 2 mL of 0.8% the FA solution. (b) Xantan gum (0.1 g), glycerin (0.4–0.5 g), and 5 mL of water were mixed with stirring until homogeneity in another beaker. (c) After the products (a) and (b) were mixed, Tween 80 (0.14 g) and melted emulsion wax (0.1 g) were sequentially introduced while heating in a water bath at 80 °C for 10 min. (d) The resulting mass was dried in an oven at 50 °C for 1 day until the film was obtained.

The excipients Kollipor® p237, Kollippor® p338, sorbitol, DMSO were introduced at the concentration indicated in Table 1 for FA release and solubilization improvement.

**Table 1.** Formulations of transdermal patches of FA. Content of FA is constant ($C_{FA}$ = 0.016 g).

| TTS | Components | Weight, g |
|---|---|---|
| Basic composition (BC) * | FA | 0.016 |
| | Xantan gum | 0.1 |
| | Glycerin | 0.5 |
| | PVP K-17 | 0.5 |
| | PEO-400 | 0.02 |
| | PEO-1500 | 0.2 |
| | Tween-80 | 0.14 |
| | Emulsion wax | 0.1 |
| | Distilled water | 7.0 |
| Kolliphor® p237 | BC + Kolliphor® p237 | 0.1 |
| Kolliphor® p338 | BC + Kolliphor® p338 | 0.085 |
| Sorbitol | BC + sorbitol | 0.4 |
| DMSO | BC + DMSO | 0.035 |

* BC is the basic composition. FA is Fulvic Acid. PVP K-17 is polyvinylpyrrolidone K-17 (poly [1-(2-oxo-1-pyrrolidinyl)ethylen]). PEO-400 is polyethylene oxide-400. PEO-1500 is polyethylene oxide-1500.

### 2.10. Release Study

The release of FA was studied according to the 2.9.4. Dissolution test for transdermal patches (Cell method by European Pharmacopoeia 11.0).

Fluorescence spectroscopy was used to evaluate FA release from the transdermal therapeutic systems. The acceptor solution (distilled water) was analyzed using the excitation spectrum, taken in the range of 300–500 nm at an emission wavelength (Em) of 520 nm. The spectrum has the peak at 360 nm and a shoulder at a wavelength of about 460 nm. The amount of the drug was determined by using a calibration curve generated from known concentrations of FA.

The penetration of FA was studied using a Franz cell with an acceptor chamber volume of 4.35 mL and 12.65 mL, respectively. An acetyl cellulose membrane (d–0.45 μm) with an area of 1.3 cm$^2$ was used for the study.

Fluorescence spectroscopy was used to evaluate the FA penetration from the transdermal therapeutic systems through the acetyl cellulose membrane. The acceptor solution (PBS at pH 7.4) was analyzed using the excitation intensity of spectra at 360 nm ($\lambda_{Em}$ = 520 nm). The receptor compartment of each cell was filled with filtered PBS (pH 7.4) and was maintained at 37 °C, whereas the surface of the membrane was left unoccluded at ambient room temperature. The receptor medium was under synchronous continuous stirring using a magnetic stirrer. Prior to applying the formulations, the diffusion cells were allowed to equilibrate for 30 min. A total of 500 μL of receptor solution was collected every 30 min for 8 h. All receptor samples were analyzed using fluorescence method. All for-

mulations were tested in three replicates, and the data are reported as mean ± standard deviation (S.D.)

The amount of the drug was determined by using the calibration curve generated from known concentrations of FA to calculate the percentage of the drug released according to Equation (1).

$$\text{Drug release (\%)} = \frac{\text{The amount of FA released at given time}}{\text{The total amount of FA loaded onto the sample}} \tag{1}$$

All the experiments were carried out in triplicate, and the data are expressed as mean ± S.D.

To assess the analytical curve for each formulation, dilutions of each compound were prepared by dissolving in water and then dilutions of each compound were analyzed with the fluorescence method. Then, the standard analytical curves were prepared. The cumulative mass of the drug permeated through area of each membrane ($\mu g/cm^2$) was plotted as a function of time. Statistical analysis of the data was performed by employing a Student's *t*-test, with the significance level set at <0.05. Data are reported as mean ± SD ($n = 3$).

### 2.11. Biological Activity

Male Wistar rats (250–300 g) were involved in the study. The animals were purchased from the Animal Breeding Facilities "Andreevka" Federal State Budgetary Institution of Science "Scientific Center for Biomedical Technologies", Federal Medical and Biological Agency (Andreevka, Moscow region, Russia). All procedures for maintenance and sacrifice (care and use) of the animals were carried out according to the criteria outlined by European Convention ET/S 129, 1986 and directives 86/609 ESC. The animals were humanely kept in plastic suspended cages and placed in a well-ventilated and hygienic rat house under suitable conditions of a room temperature (27 ± 2 °C) and humidity. They were given food and water ad libitum and subjected to a natural photoperiod of 12 h light and 12 h dark cycle. The animals were allowed two weeks of acclimatization prior to the animal model experiments in the study.

### 2.11.1. Model of Adjuvant-Induced Arthritis

The study was conducted in accordance with the Declaration of Helsinki and approved by the Local Ethics Committee of Privolzhsky Research Medical University, Russian Federation (protocol No. 2 from 17 February 2023).

The animals were deeply anesthetized on intraperitoneal administration of Zoletil 100 (60 mg/kg) and Xyl (6 mg/kg). The skin covering the knee joint was incised and the patellar tendon was exposed. A total of 0.2 mL of Complete Ferund's Adjuvant containing 0.2 mg of Mycobacterium tuberculosis suspended in 0.17 mL of paraffin oil sterilized (Erba Lachema, Brno, Czech Republic) was injected through the patellar tendon into the right knee joint cavity to induce pain and inflammation (day 0) [34]. Rats were returned to the home cage, allowed to recover, and monitored daily for gait disturbances.

Fifteen days after Complete Ferund's Adjuvant injection, the animals were divided into three equal groups such that the cross-sectional diameter of the knee cartilage was equivalent. The joint diameter was measured by a micrometer. The arthritic joint when compressed with a micrometer elicited pain that was indicated by squeaking and paw withdrawal. Patches were applied to the right hind paw in the knee joint area for 8 h two times in day per day for 7 days. Doses were standardized by applying patches measuring 1.0 cm × 0.84 cm in size, based on the difference between human body weight and rat body weight. The group of rats that did not applied patches (without treatment) was used as the control group.

### 2.11.2. Histopathological Studies

Samples of the right knee joint of rats were removed and fixed in 10% neutral phosphate-buffered formalin for 48 h. After this process, they were decalcified for 2–5 days in a 5–8% buffered formic acid solution with a daily change of the decalcifying solution and

monitoring the completeness of decalcification. The samples were dehydrated in a series of ethanol from 70 to 99.9%, purified in xylene, embedded in paraffin blocks, and subjected to a 4 μm thick cross section. Staining was performed with hematoxylin-eosin and Masson's trichrome, examined and photographed using a photon microscope (Leica DM1000; Leica Microsystems, Germany). Histological examination of the knee joint included assessment of chondrocyte degeneration, cartilage erosion/ulceration, cartilage fibrillation, mononuclear cell infiltration of the synovium (–normal, +minimal, ++mild, +++moderate, ++++severe).

### 2.11.3. Evaluation of Biochemical Indexes

Biochemical indexes were obtained using blood on the seventh day of treatment. The blood was stabilized with sodium citrate (1:9). Erythrocytes were washed twice with 0.9% NaCl by centrifugation for 10 min at $1600 \times g$. The intensity of lipid peroxidation was estimated by the malondialdehyde (MDA) level in plasma and erythrocytes in accordance with Uchiyama and Mihara methods [37]. Superoxide dismutase activity (SOD) (EC 1.15.1.1) was measured in erythrocytes using inhibition of adrenaline auto-oxidation [38]. Catalase activity (EC 1.11.1.6) was determined spectrophotometrically based on the decomposition of hydrogen peroxide by the catalase [39]. Glutathione reductase activity (GR) (EC 1.8.1.7) was studied by spectrophotometry based on the oxidized glutathione reduction [40]. The energy metabolism in erythrocytes was studied using the catalytic activity of lactate dehydrogenase (EC 1.1.1.27) in direct (LDH direct, substrate—50 mM sodium lactate) and reverse (LDH reverse, substrate—23 mM sodium pyruvate) reactions [41]. The specific activity of the enzymes was calculated using the protein concentration analyzed by the modified Lowry method [42]. The activity of aldehyde dehydrogenase (ALDH) (EC 1.2.1.3) was estimated spectrophotometrically in accordance with the previous methods [43].

To assess the intensity of the inflammatory process, the total number of leukocytes in the blood was determined using the automatic analyzer BC-2800 Vet (Mindray, Shenzhen, China) and the concentration of highly sensitive C-reactive protein (Rat CRP ELISA Kit, Elabscience Biotechnology, Wuhan, China). The content of rheumatoid factor (RF) was determined in the blood serum of rats by the turbidimetric method using Cobas 6000 ("F. Hoffmann-LaRoche Ltd.", Basel, Switzerland) and CobasIntegra 400 ("F. Hoffmann-LaRoche Ltd.", Basel, Switzerland) devices [44]. The erythrocyte sedimentation rate (ESR) of rat blood was determined by the Panchenkov method [45].

### 2.12. Statistical Analysis

Statistical data were processed by the software, Statistica 6 (StatSoft Inc., Tulsa, OK, USA). The normality of the results distribution was shown using the Shapiro–Wilk test. The significance of differences between groups was assessed using Student's *t*-test and one-way analysis of variance (ANOVA). The differences were considered statistically significant at $p < 0.05$.

## 3. Results

### 3.1. FA Properties and Transdermal Patches Preparation

The final orange-colored product (FA) extracted from the peat had the following properties: (a) FTIR, ν, $cm^{-1}$: 3394 (OH— hydroxyl in carboxyl, alcohol and phenolic groups), 2937, 2920, 2850 ($CH_3$—, $CH_2$—, —CH—), 1716 (C=O in carbonyl groups), 1608 (C=O in carboxyl groups), 1280 (C-O in phenolic hydroxyl), (Figure S1a); (b) solid-state $^{13}$C NMR, δ, ppm for carbon atoms: 17.84 and 29.27 (aliphatic), 71.86 (hemiacetal), 116.02 (olefinic), 130.24 (aromatic), 157.00 (O-substituted), 173.90 (carboxyl), 198.77 (carbonyl) (Figure S1b). Assay of carboxyl groups ([COOH] = 7.1 mmol-eq/g) and phenolic groups ([Ph-OH] = 4.8 mmol-eq/g) was carried out by the methods [46,47]. The appearance of the crystals, assessed by optical microscopy, SEM, and energy dispersive X-ray spectrum (EDX analysis), was presented in Figure S2. In distilled water, the zeta potential of FA particles in the 0.02% solution at pH 2–3 corresponded to −27.9 mV, the particle size was 8–10 nm.

Figure 1 shows the general FA formula and some characteristics:

**Figure 1.** The general FA formula. $C_{34}H_{30}O_{18}$; C:O = 56.2%:39.6%; M.m. = 726 g/mol (calculated); M.m. = 740 g/mol (determined by cryoscopy; solvent–water); logP = 2.93 +/− 1.29 (calculated). The FA solubility was 3.3 mL of water per gram (freely soluble).

FA assay in its pure substance and in the dosage form was carried out by spectrofluorimetry using excitation spectra, which are characterized by the more symmetrical band, better limit of detection (LOD = 0.2 ppm (μg/mL)), and lower limit of quantitation (LLOQ = 0.5 ppm (μg/mL)) compared to emission spectra (Figure 2).

Figure 2 and Table S1 show data confirming the linear nature of the calibration curve ($R^2$ = 0.9934). The linearity of the developed method was proven within the limits of compound concentrations of 1–10 μg/mL. The linear regression equation was Y = aX + b, where a = 67.309, b = 0.0024.

The calculated RSD coefficients of variation indicate that the proposed method for the quantitative determination of FA by measuring the fluorescence of solutions satisfies the precision index.

To design the transdermal patch and optimize its composition, it is necessary to evaluate the solubility of the active substance, depending on its hydrophilicity or hydrophobicity. We have shown that the solubility of fulvic acid in water is 3.3 mL of water per gram, which corresponds with the term "freely soluble" and characterizes FA as a polar hydrophilic molecule.

As a combination of xanthan gum and low molecular (15,000–20,000 g/mol) PVP, emulsion wax used as a lipid component, and glycerin, PEO, Tween 80 were used as surfactants, plasticizers, and transcutants. The formulations are given in Table 1. Moreover, Kolliphor® poloxamers, DMSO, and sorbitol were used to improve the solubilization of polar hydrophilic FA.

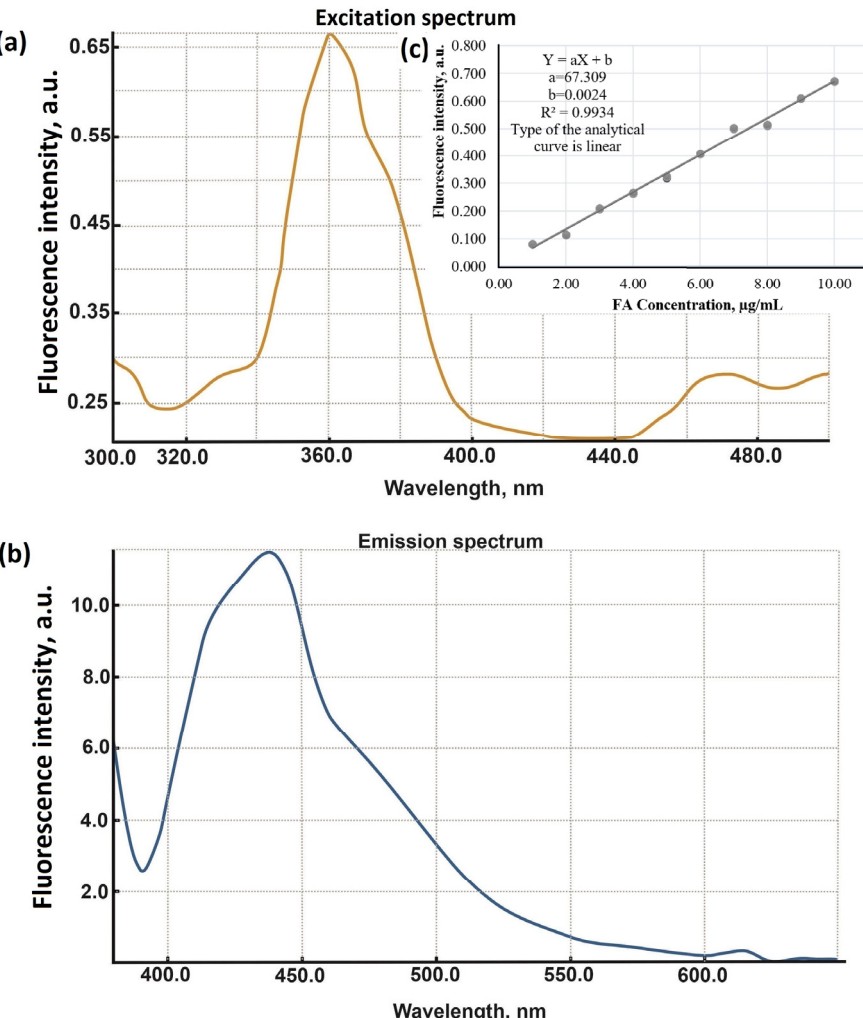

**Figure 2.** Fluorescence spectra (**a**,**b**) of 0.001% FA aqueous solution and analytical curve (**c**). One-dimensional emission spectra were taken in the wavelength range of 400–550 nm at a constant excitation wavelength of 360 nm ($\lambda$ex = 360 nm). Excitation spectra were recorded in the range of 300–500 nm at a fixed emission wavelength of 520 nm ($\lambda$em = 520 nm).

### 3.2. The Release of FA through the Cellulose and Acetyl Cellulose Membranes

Table 2 and the Figure 3 show the release study dates of FA through the cellulose membrane from transdermal patches. The results were compared with the basic composition.

**Table 2.** The results of FA release from the patches through the regenerated cellulose membrane.

| Curve Color | Patch | $C_{FA}$, mg·mL$^{-1}$ | Release, % | Plateau Time, h |
|:---:|:---:|:---:|:---:|:---:|
| | Kolliphor® p237 | 8.1 ± 0.2 | 56.2 | 8.25 |
| | DMSO | 6.7 ± 0.2 | 46.5 | 5.0 |
| | Kolliphor® p338 | 4.6 ± 0.3 | 31.9 | 8.5 |
| | Basic composition | 4.8 ± 0.2 | 33.3 | 6.5 |
| | Sorbitol | 4.0 ± 0.2 | 27.8 | 10.0 |

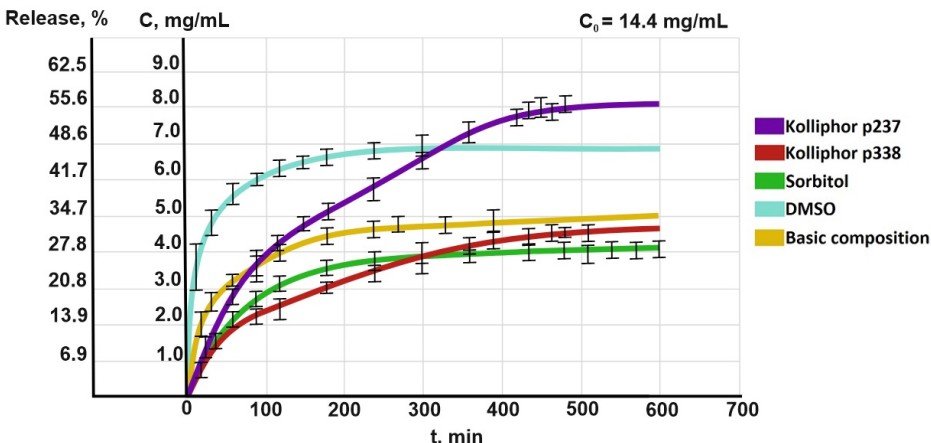

**Figure 3.** Time dependence of FA release from the transdermal patches through the regenerated cellulose membrane. FA concentration was estimated by fluorescence excitation spectra at 360 nm.

It was shown that the patches with sorbitol and Kolliphor® p338 did not affect the release rate. The release rate of FA from a DMSO-containing transdermal patch at the initial stage (up to 250 min) is the maximum, but the release percentage reached only 45%. The closest result to this time was obtained for the patch with Kolliphor® p237. With a longer time (500–600 min), the degree of FA release from this patch reached 56%.

We simulated the dynamics of the FA penetration from the patches on a model of acetyl cellulose membrane using a vertical Franz cell (pH of the media is 7.4). We studied the release of FA only from the patch with Kolliphor® p237 and compared with the basic composition. Since DMSO (the internal positive standard in the previous experiment) can cause irritation and allergic reactions and side effects, its effect was not investigated. Figure 4 show the results of the FA release for the patches.

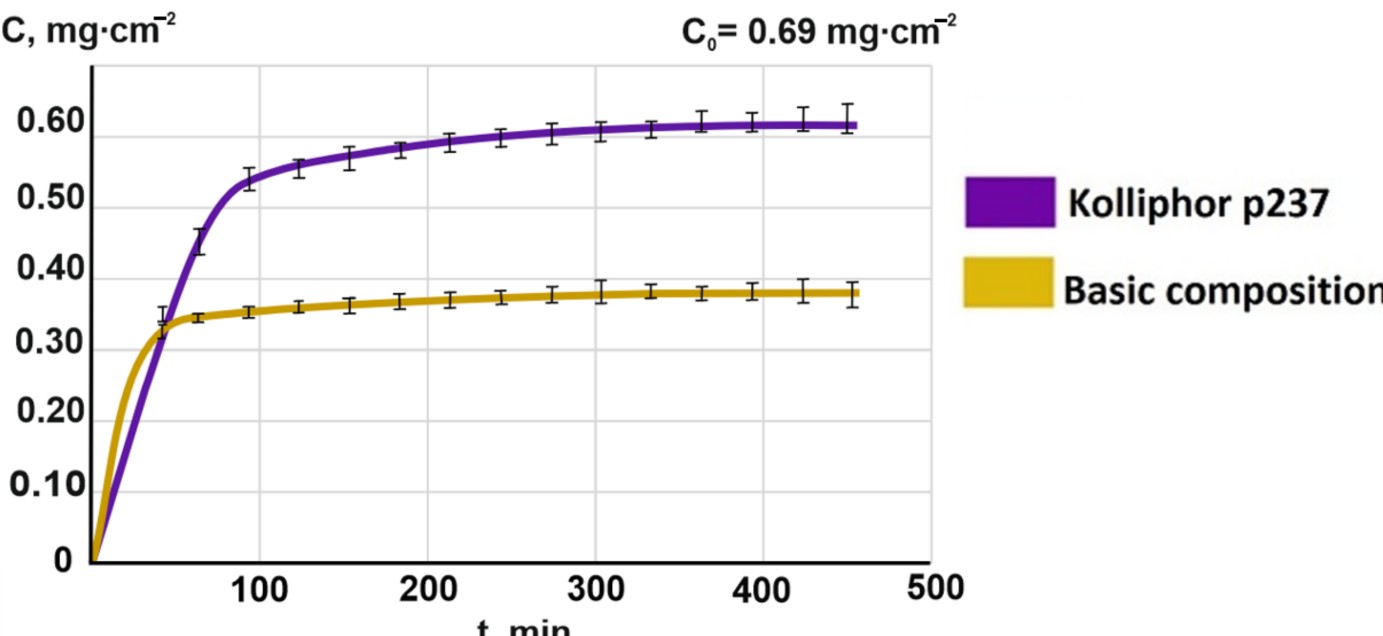

**Figure 4.** Time dependence of FA release from the transdermal patches through the acetyl cellulose membrane. FA concentration was estimated by fluorescence excitation spectra at 360 nm.

The initial surface concentrations of FA in the patches on an acetyl cellulose membrane with an area of 1.3 cm$^2$ were 0.69 mg/cm$^2$. The concentration values at the plateau for 8 h was equal to $0.62 \pm 0.02$ for Kolliphor® p237 and $0.38 \pm 0.02$ for basic composition, respectively.

Thus, we have designed the transdermal patch with fulvic acid as a potential API. It is shown that the Kolliphor® p237 patch is the most optimal in term of release.

### 3.3. Biological Activity of the FA Patch on the Adjuvant-Induced Arthritis Model in Rats

A few hours after the injection, the limbs injected with Freund's complete adjuvant were swollen and significantly enlarged. Three weeks after the injection, the diameter of the knee joint visually increased significantly by 120–130%. This marked increase in joint diameter confirmed the successful development of inflammation. The development of arthritis ended on the 15th day after the introduction of complete Freund's adjuvant into the knee joint, after which treatment was started using a patch for 7 days.

Results of histopathological study of complete adjuvant-induced arthritis in Wistar rats are shown in Figure 5 and Table 3.

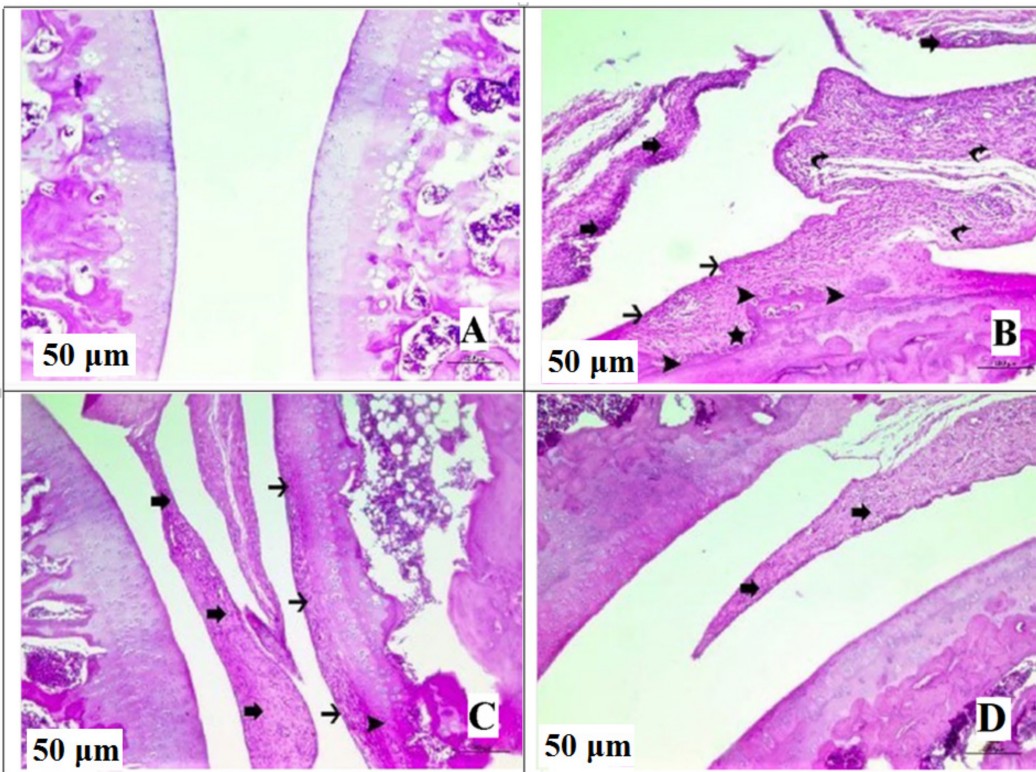

**Figure 5.** Histopathological results (H&E, ×100 magnification, scale bar 50 μm). (**A**) Intact group, (**B**) Control group (without treatment), (**C**) Experimental group (basic composition), (**D**) Experimental group (Kolliphor® p237). Black thin arrow, arrowhead, and asterisks indicate cartilage fibrillation, erosion, and ulceration, respectively. A thick arrow indicates synovial infiltration with mononuclear cells. The curved arrow indicates the formation of the Pannus.

**Table 3.** Histopathological study of adjuvant-induced arthritis.

| Groups | Chondrocyte Degeneration | Erosion/Ulceration of Cartilage | Cartilage Fibrillation | Infiltration |
|---|---|---|---|---|
| Intact | − | − | − | − |
| Control (without treatment) | ++++ | ++++ | ++++ | ++++ |
| Basic composition | ++ | ++ | ++ | +++ |
| Kolliphor® p237 | + | + | ++ | ++ |

Severity: − No; + Minimum; ++ Light; +++ Moderate; ++++ Heavy.

The normal histological structure of the joint capsule, synovial membrane, cartilage, and bone tissue was observed in the intact group. Conversely, the degeneration of chondrocytes, cartilage erosion/ulceration, cartilage fibrillation, and infiltration of the synovial membrane by mononuclear cells were observed in the control group (without treatment). In the experimental groups (basic composition, Kolliphor® p237), the degenerative processes decreased (Table 3).

The group treated with Kolliphor® p237 patch had minimal degeneration of chondrocytes and cartilage erosion, as well as moderate fibrillation of the cartilage and infiltration of the synovial membrane by mononuclear cells.

This result may indicate that Kolliphor® p237 patch had an anti-inflammatory effect. Table 4 presents the data on the specific antioxidant activity in rats.

**Table 4.** Level of antioxidant activity in rats ($n = 3$).

| N | Groups | SOD, inh/min·mg Protein (%) | Catalase, μmol $H_2O_2$/min·mg Protein (%) | GR, NADPH/min·mg Protein (%) | G6PD, NADPH/min·mg Protein (%) | LDH dir., NADPH/min·mg Protein (%) | LDH rev., NADPH/min·mg Protein (%) |
|---|---|---|---|---|---|---|---|
| 1 | Intact | 1013.0 ± 61.2 (100%) | 34.7 ± 1.7 (100%) | 87.4 ± 3.1 (100%) | 43.2 ± 2.9 (100%) | 42.2 ± 2.4 (100%) | 183.7 ± 7.5 (100%) |
| 2 | Control (without treatment), τ = 15 days | 1502.0 ± 39.2 (148%) | 58.9 ± 2.0 (170%) | 131.1 ± 3.8 (150%) | 64.2 ± 2.9 (149%) | 73.3 ± 5.3 (174%) | 274.8 ± 8.9 (150%) |
| 3 | Control (without treatment), τ = 21 days | 805.0 ± 37.4 (79%) | 33.2 ± 0.6 (96%) | 120.9 ± 5.9 (138%) | 60.1 ± 1.8 (139%) | 65.5 ± 4.1 (155%) | 251.6 ± 7.8 (137%) |
| 4 | Basic composition (7 days of treatment) | 997.4 ± 20.8 (98%) | 42.0 ± 1.8 (121%) | 102.5 ± 3.6 (117%) | 62.6 ± 3.0 (118%) | 53.2 ± 4.2 (126%) | 226.0 ± 13.7 (123%) |
| 5 | Kolliphor® p237 (7 days of treatment) | 1174.2 ± 43.1 (116%) | 44.5 ± 1.5 (128%) | 94.2 ± 3.2 (108%) | 56.0 ± 3.5 (123%) | 51.0 ± 3.0 (121%) | 218.8 ± 5.4 (119%) |

Table values are mean values + dispersion (Mann–Whitney test, $p < 0.05$); values are statistically significant in comparison with the intact group (Mann–Whitney test, $p < 0.05$). SOD is superoxide dismutase. GR is glutathione reductase. G6PD is glucose-6-phosphate dehydrogenase. LDH dir. is catalytic activity of lactate dehydrogenase in direct reactions. LDH rev. is catalytic activity of lactate dehydrogenase in reverse reactions.

On the 15th day after the resulting inflammation in the absence of treatment, the specific activity of all antioxidant defense enzymes (SOD, catalase, GR, G6PD, LDR directed and LDR reversed) increased, which characterizes their activation as a response to oxidative stress developing against the background of RA. These results are consistent with the studies of the authors, which demonstrated a significant increase in enzyme activity in various tissues (liver, heart, muscles, blood serum), from 1.2 to 1.7 times [48–52].

Along with the development of oxidative stress against the background of RA on the 15th day of modeling, the RA process is also evidenced by an increase in the specific activity of ALDH and the level of MDA both in plasma and in erythrocytes (Table 5). After 7 days of treatment (from days 15 to 21 of the experiment) with transdermal patches with the basic composition and the composition named Kolliphor® p237, the level of specific activity of all enzymes returned to normal and was close to that of the group of intact animals (positive control) (Table 4). A slight increase in the level of specific activity of SOD, catalase, GR, and G6PD was observed after treatment with transdermal patches, which is probably due to the antioxidant properties of FA.

The activity of ALDH in the treatment of both the basic composition and the Kolliphor® p237 composition decreased compared to the control without treatment toward the normalization of this indicator. The level of MDA in erythrocytes under the action of FA in the transdermal patch also decreased. It should be noted that MDA in erythrocytes is reduced almost to the state of intact animals (Table 5).

The positive dynamics toward the normalization of biochemical indexes reflecting oxidative stress is probably due to a decrease in the load on the studied enzyme systems of antioxidant protection under the influence of FA, as well as to a decrease in lipid peroxidation processes due to their antioxidant properties.

**Table 5.** Biochemical indexes of LPO in rats ($n = 3$).

| N | Groups | ALDH, nmol NADPH/min·mg Protein (%) | MDA (in Plasma), nmol NADPH/min·mg Protein (%) | MDA (in Erythrocytes), nmol NADPH/min·mg Protein (%) |
|---|---|---|---|---|
| 1 | Intact | $41.9 \pm 3.1$ (100%) | $0.9 \pm 0.0$ (100%) | $7.3 \pm 0.3$ (100%) |
| 2 | Control (without treatment), $\tau = 15$ days | $65.7 \pm 5.2$ (157%) | $2.5 \pm 0.1$ (278%) | $12.3 \pm 0.2$ (168%) |
| 3 | Control (without treatment), $\tau = 21$ days | $61.6 \pm 4.8$ (147%) | $2.2 \pm 0.1$ (242%) | $11.9 \pm 0.2$ (164%) |
| 4 | Basic composition (7 days of treatment) | $54.7 \pm 3.4$ (131%) | $1.8 \pm 0.0$ (204%) | $8.7 \pm 0.3$ (119%) |
| 5 | Kolliphor® p237 (7 days of treatment) | $51.6 \pm 1.4$ (123%) | $1.7 \pm 0.1$ (191%) | $7.9 \pm 0.2$ (108%) |

Table values are mean values + dispersion (Mann–Whitney test, $p < 0.05$); values are statistically significant in comparison with the intact group (Mann–Whitney test, $p < 0.05$). ALDH is aldehyde dehydrogenase. MDA is malondialdehyde.

More objective indicators of reducing the inflammatory process in the treatment with transdermal patches are indicators of inflammation markers (ESR, white blood cell count, C-reactive protein, RF). On the 15th day, with the development of the inflammatory process, the level of inflammatory markers increased sharply (Table 6). In the course of treatment from 15 to 21 days with the transdermal patches with FA, the following indicators decreased: ESR from 10.5 to 4.9–3.8 mm/h, the number of leukocytes from 19.4 to 11.6–10.4 (109/L), the concentration of C-reactive protein from 12.6 to 5.0–4.5 ng/mL, and RF concentration from 8.2 to 6.3–5.0 IU/mL. On the 21st day without treatment, these indicators were unsatisfactory (Table 6).

**Table 6.** Level of inflammatory markers in rats ($n = 3$).

| N | Groups | ESR, mm/h | Leukocytes, $10^{-9}$/L | C-Reactive Protein, ng/mL | RF, IU/mL |
|---|---|---|---|---|---|
| 1 | Intact | $3.5 \pm 0.2$ (100%) | $9.9 \pm 0.5$ (100%) | $4.0 \pm 0.4$ (100%) | $5.4 \pm 0.3$ (100%) |
| 2 | Control (without treatment), $\tau = 15$ days | $10.5 \pm 0.5$ (300%) | $19.4 \pm 1.0$ (196%) | $12.6 \pm 0.6$ (315%) | $8.2 \pm 0.2$ (152%) |
| 3 | Control (without treatment), $\tau = 21$ days | $8.6 \pm 0.3$ (246%) | $16.6 \pm 1.4$ (168%) | $10.7 \pm 0.8$ (267%) | $7.8 \pm 0.4$ (144%) |
| 4 | Basic composition (7 days of treatment) | $4.9 \pm 0.2$ (140%) | $11.6 \pm 0.6$ (117%) | $5.0 \pm 0.7$ (125%) | $6.3 \pm 0.2$ (117%) |
| 5 | Kolliphor® p237 (7 days of treatment) | $3.8 \pm 0.4$ (109%) | $10.4 \pm 0.7$ (105%) | $4.5 \pm 0.6$ (112%) | $5.0 \pm 0.3$ (93%) |

Table values are mean values + dispersion (Mann–Whitney test, $p < 0.05$); values are statistically significant in comparison with the intact group (Mann–Whitney test, $p < 0.05$). ESR is the erythrocyte sedimentation rate. RF is the rheumatoid factor.

In general, it should be noted that all the analyzed indicators and markers of inflammation in the course of treatment for 21 days practically returned to normal.

## 4. Discussion

FA isolated from the peat in the Nizhny Novgorod region (Russia) may be a potential API for the preparation of the transdermal patch. The product (FA) obtained by us was characterized as a pure substance, close in properties to the literature data for this group of compounds [53–55]. We carried out screening studies on the stability of the original substance-FA both in aqueous solutions and in the patch composition. When stored in air at room temperature for three months, the assay of FA in the aqueous media remained virtually unchanged. However, during storage over the same period, the FA aggregation changed and the particle size increased from 8 nm to 32 nm. It has been shown that FA oxidation does not occur, and aggregation is insignificant during the preparation time of the emulsion base.

Formulations of the anti-inflammatory transdermal patches with fulvic acid on the emulsion basis were proposed due to the suggestion that FA dissolves colloidally in the hydrophobic part of the emulsion. For optimal solubilization and effective penetration of FA into the stratum corneum, it is necessary to introduce the pluronic into the composition; the paper shows the advantages of using the pluronic Kolliphor® p237 at the concentration of at least 1%. The emulsion hydrophilic base for the transdermal patches allows FA to

accumulate in the stratum corneum when it is released from the base, creating a kind of depot. In addition, emulsion bases are more stable than the easily oxidized lipophilic bases and tend to last longer [56–59].

In vitro studies of the release of FA from the transdermal patch using the Franz diffusion cell showed its high efficiency (56% and 90%, respectively, in 8 h). The effectiveness of the action of the transdermal patch with Kolliphor® p237 is probably due to the higher solubilizing ability of supramolecular nanosized FA particles formed in solution compared to Kolliphor® p338. In addition, the decrease in FA release from the Kolliphor® p338 patch is likely due to its thermoreversing properties. Molecules of Kolliphor® p338 form a rigid hydrophobic matrix at an elevated temperature, and its destruction occurs at a lower temperature. Such properties lead to the decrease in the solubilization of FA.

The pluronic (Kolliphor® p237), which we used in the emulsion base (oil-in-water emulsion) of the transdermal patch, is capable of forming a nanosized delivery system. The oil particles are capable of immobilizing FA molecules with dimensions of about 8 nm. In further studies, it is probably possible to consider some Pluronics as components of a nanoscale delivery system for the treatment of rheumatoid arthritis [60].

The use of FA against the RA pathology led to the normalization of marker indicators of oxidative stress toward the control values (Tables 3–5), which is apparently due to the antioxidant properties of FA. There was the normalization of the specific activity of the enzyme unit (SOD, catalase), as well as the normalization of the activity of some enzymes of oxidative metabolism (G6PD, GR). It was shown that, at the same time, the indicators of lipid peroxidation (MDA, ALDH) decreased. In addition, there was an improvement in the severity of the disease and the intensity of inflammation (Figure 5, Table 6). Improved treatment rates for RA are associated not only with the antioxidant properties of FA, but also, probably, with the inhibition of the transcription factor Nf-kB, as shown in a number of studies [61–63].

Transdermal administration of FA, which has diverse biological activities, is an attractive alternative to oral medications and injections in the treatment of rheumatoid arthritis (RA). Transdermal patches avoid the negative effects of cyclooxygenase (COX) inhibitors and minimize gastrointestinal side effects [32]. The patches are easy to apply to the skin and do not require repeated applications like gels. This makes them more convenient for patients.

## 5. Conclusions

For the first, the composition of the transdermal patch with FA has been proposed for the treatment of anti-inflammatory diseases using RA as an example. Pluronic Kolliphor® p237 showed the transcutant property of FA in the transdermal patch on the water-emulsion base improving FA release. The efficacy of the FA patch has been demonstrated in in vivo experiments in rats in the arthritis model using complete Freund's adjuvant. The influence of FA against the background of pathology was manifested in the change of the marker parameters toward the positive control (intact rats). Fulvic acid in the transdermal patch affects the antioxidant defense enzyme system, normalizing biochemical indexes such as SOD, catalase, GR, G6PD, LDR directed, and LDR reversed in the experiment on rats with inflammation.

Thus, it can be assumed that the use of the transdermal FA patches can be effective in the treatment of various inflammatory diseases triggered by oxidative stress.

**Supplementary Materials:** The following supporting information can be downloaded at: https://www.mdpi.com/article/10.3390/scipharm91040045/s1, Figure S1: FTIR spectrum (a) and Solid-state [13]C NMR spectrum (b) of FA; Figure S2: SEM images (a–d), EDX spectra and the distribution of elements in powders on aluminum foil (e) and FA appearance (f) respectively; Table S1: Standard deviations of the results of measurements of the fluorescence of the FA solution.

**Author Contributions:** N.B.M. conceived and designed the experiments, analyzed the data, and wrote the paper; A.G.S. and P.V.P. performed the biological experiments, M.A.K. and A.A.V. obtained the substance, formulations, and performed the physicochemical experiments. All authors have read and agreed to the published version of the manuscript.

**Funding:** This research received no external funding.

**Institutional Review Board Statement:** The study was conducted in accordance with the Declaration of Helsinki and approved by the Local Ethics Committee of Privolzhsky Research Medical University, Russian Federation (protocol No. 2 from 17 February 2023).

**Informed Consent Statement:** Informed Consent Statement: Informed consent was obtained from all subjects involved in the study.

**Data Availability Statement:** The data presented in this study are available upon reasonable request from the corresponding author.

**Acknowledgments:** The SEM: EDX and BET studies were carried out on the equipment of the Collective Usage Center "New Materials and Resource-saving Technologies" (Lobachevsky State University of Nizhniy Novgorod).

**Conflicts of Interest:** The authors declare no conflict of interest.

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
