# Peer review of "Anti-Inflammatory Property Establishment of Fulvic Acid Transdermal Patch in Animal Model"

_scipharm, doi:10.3390/scipharm91040045_

Round 1

Reviewer 1 Report

Anti-inflammatory Transdermal Patch with Fulvic Acid. Design and Study in the Adjuvant-Induced Arthritis in Rats

The submitted manuscript is dealing with the investigation on the preparation of an anti-inflammatory transdermal patch with fulvic acid (FA) based on an emulsion basis using Kolliphor® p237 as a permeability enhancer was developed and studied. FA was isolated from the peat in the Nizhny Novgorod region of Russia and characterized as a potential active pharmaceutical ingredient. In vitro studies of the release of FA from the transdermal patch, as well as the permeability of the patch using the Franz diffusion cell, showed its high efficiency (56% and 90%, respectively, in 8 hours). In the in vivo experiment, qualitative and quantitative features of the rat knee caused by complete Freund’s adjuvant-induced arthritis (morphological changes, the FA influence on the biochemical indexes) were studied. The results presented are interesting and support the potential of using transdermal patch with fulvic acid for arthritis cancer treatment. However, paper needs corrections prior to publication.

Major comments

- Regarding the in vivo experiment, there is no explanation about the patch application on the rat knees, it needs some clarification, please indicate which Ethical Committee was followed.

- Do you consider carrying on stability studies to complete this work?

- References. Kindly elaborate more on the sub-sections of the Results and Discussion section with references. Appropriate references are required to support the claim and results of any review.

Minor comments

- Please reorganize the Introduction section, putting the aim of the work at the end.

- Please state the chemical names of all excipients together with the commercial names. Consider that Kolliphor® is a synonym of Pluronic®.

- Please, layout your publication in an appropriate way, paying attention to the margins and framing of the figures, for example, page 6, figure 1.

- Figure 1: kindly use a large font while making images. Please, identify better the calibration curve.

- Please explain all abbreviations before using them, in order to better and quickly follow the text (ROS, ESR, RF, SOD, G6PD, LDH, MDA, ADH, PVP K-17, PEO-400, PEO-1500, DMSO).

- Please, write in vivo and in vitro with cursive letters when used throughout the text, including title of sub-sections, and figures.

- I would remove the pH analysis from the Methods section.

- I would remove lines 148-158, page 4 from Materials and Methods section, and I would include them in the Results and Discussion section.

- The conclusion part of the manuscript is too lengthy. Please organize the conclusion in a short and precise way in a single paragraph. 

Please correct these typing errors:

 - Page 3, line 96: Please, correct twin-80.

- Page 3, line 101: Please, correct 13C NMR.

- Page 3, line 118: Please, correct cm−1.

- Page 3, line 143: Please, correct H2SO4.

- Page 4, line 164: Please, correct 80 oC.

- Page 4, line 166: Please, correct Kollipor p237, Kollippor p338.

- Page 6, figure 1: Please, correct Excication spectrum.

Reviewer 2 Report

The present article evaluates the implications of fulvic acid as a transdermal patch in ameliorating inflammatory processes in adjuvant-induced arthritis in rats. The subject is promising and complex approached but requires major changes in form and content:

1. It is preferable that the title NOT be split into two sentences

2. The abstract needs changes in the introductory background (introduction of the evaluated topic-first sentences) and in the conclusion of the abstract which should point to future outcomes and to what extent they can be applied to patients.

3. L36 - either mention other properties or remove 'other properties' - need to be more specific in scientific articles

4. L35-36 - there are too many bibliographic resources for such a small amount of information - I recommend selecting the most current and complex ones

5. Given the unmet needs due to the inability of certain pharmaceutical forms to reach the target site in rheumatoid arthritis, it is advisable to specify the role of nanomedicine in addressing these unmet needs in arthritis, as well as the possibilities of incorporating fulvic acid into nanosystems to deliver the target. I suggest checking and referring to: PMID: 37031724.

6. Abbreviations used in Tables 1, 4, 5, 6 should be explained in the table legend.

7. If possible, it would be advisable to increase the clarity of Figure 1(a).

8. It is imperative that the discussion section be separately created and designed to address in detail and evaluate the results obtained in relation to other similar studies, as well as to present in the last part the main strengths and limitations that can be addressed by future research directions.

9. The conclusions section should be shortened and addressed more specifically explaining to what extent the results of this study can be adapted for administration to patients.

Reviewer 3 Report

the authors prepared a transdermal patch of fulvic acid, characterised it and performed in vitro and in vivo analyses to demonstrate its anti-inflammatory activity. I have several questions:

Fulvic acid is derived from peat. I have seen the spectra. What other components are in peat? What is the solubility, pKa and logP of fulvic acid?

Why these components were chosen for the peat, please discus these more. What is the role of DMSO in the composition? Have biocompatibility studies been done on the formulation? If not why not?

The results are not well-discussed, the conclusion is too short, the novelty needs to be better explained and supplemented by the answers to my questions.

Moderate english editing is needed.

Reviewer 4 Report

Dear Authors,

I write you in regard to your manuscript entitled "Anti-inflammatory Transdermal Patch with Fulvic Acid. Design and Study in the Adjuvant-Induced Arthritis in Rats". 

- Abstract must be revised and rewritten. Reading did not flow along the text.

- the vertical diffusion cell test had a synthetic membrane that can not be considered a model of stratum corneum. Please, revise this part.

- the correct term is analytical curve.

- please, even in a supplementary material, add some data about the analytical method.

- it was not clear if the permeability assay was in the synthetic membrane. What was the model membrane?

Round 2

Reviewer 1 Report

I recommend this revised article for being published in the Sci Pharm.

Author Response

Dear Reviewer, thank you very much for your comment!

Reviewer 2 Report

The authors have significantly improved the manuscript based on the suggestions received.

Author Response

(The authors gave the same response as above.)

Reviewer 3 Report

The authors replied my questions.

Minor English editing is required.

Author Response

(The authors gave the same response as above.)

Reviewer 4 Report

Dear Authors,

Thank you immensely for addressing the peer-review questions.

- please, the title must be revised, for instance: Anti-inflammatory Property Establishment of Fulvic Acid Transdermal Patch in Animal Model. 

-Abstract seemed to still need revision.

- Please, add reference(s) in line 70.

- Considering that your membrane model was the synthetic one, please, try to avoid the term "permeability". It was performed a release test.

- If by calibration plot you meant to analytical curve, please, provide correction (line 197).

- lines 498-502 were too speculative and it is not acceptable such type of text in the conclusions. Try just to answer your objectives in your conclusion section.

- Try to update your references of MDA, by consulting: doi:10.1016/j.ab.2012.01.016; and DOI: 10.1111/ics.12874 

Author Response

Dear Reviewer, thank you so much for your comments! Please, see the attachment
